# Microclimate and Vegetation Structure Significantly Affect Butterfly Assemblages in a Tropical Dry Forest

Anirban Mahata [1], Rajendra Mohan Panda [2], Padmanava Dash [3,*], Ayusmita Naik [1], Alok Kumar Naik [1] and Sharat Kumar Palita [1,*]

[1] Department of Biodiversity and Conservation of Natural Resources, Central University of Odisha, Koraput 764021, India
[2] Department of Integrative Biology, University of South Florida, Tampa, FL 33613, USA
[3] Department of Geosciences, Mississippi State University, Mississippi State, MS 39762, USA
* Correspondence: pd175@msstate.edu (P.D.); skpalita@cuo.ac.in (S.K.P.)

**Abstract:** Understanding the factors that influence the diversity and distribution of butterfly species is crucial for prioritizing conservation. The Eastern Ghats of India is an ideal site for such a study, where butterfly diversity studies have yet to receive much attention. This study emphasized the butterfly assemblages of three prominent habitats in the region: open forests, riparian forests, and dense forests. We hypothesized that riparian forests would be the most preferred habitat for the butterflies, as they provide suitable microclimatic conditions for butterflies. The study collected samples for 35 grids of $2 \times 2$ km$^2$ for each habitat during the dry months (December–June). We considered the relative humidity, temperature, light intensity, elevation, and canopy cover to assess their influences on butterfly richness and abundance. We also considered the impact of disturbances on their distribution. We used structural equation modeling and canonical correspondence analysis to quantify the correlation and causation between the butterflies and their environment. The study recorded 1614 individual butterflies of 79 species from 57 genera and 6 families. During the study, we found that temperature was the most significant factor influencing butterfly richness. Relative humidity was also important and had a positive impact on butterfly richness. Riparian forests, where daytime temperatures are relatively low, were the most preferred microhabitat for butterflies. Open forests had greater species diversity, indicating the critical significance of an open canopy for butterflies. Though riparian forests need greater attention concerning butterfly distribution, maintaining open and dense forests are crucial for preserving butterfly diversity.

**Keywords:** diversity; distribution; India; microhabitat; riparian forests





## 1. Introduction

Understanding the diversity and distribution of a species' composition is crucial for prioritizing its conservation. The microclimate, which is defined as the immediate environment of a species, shapes the distribution and survival of a species. At the community level, the microclimate constrains the evolution of a species' life-history strategies and niche segregation, as well as their persistence [1]. The microclimate is closely linked to the vegetation structure and is a critical factor in controlling a species' distribution in microhabitats [2]. The forest microclimate is an important component of habitat quality, influencing species' survival, reproductive success, and behavior [3,4]. A suitable microclimate that influences the microhabitat, diapause (i.e., larval growth), and food availability is also highly essential for the survival of butterfly species [5]. However, the significance of the microclimate on butterfly richness and distribution requires greater emphasis [6,7]. Several studies have quantified the differences in the vegetation structure among microhabitats [2,6–8] and their influence on host specificity and the distribution of butterflies [8,9]. Riparian forests act as ecological conduits for the wildlife and ecotones of different habitat fragments [10].

Open forests harbor species that are tolerant of habitats that have experienced disturbances in their conditions, and they show greater species diversity than dense forests with low understory vegetation [11]. Butterflies are sensitive to habitat conditions and show quick responses to small changes in their environmental set-ups. In particular, butterflies are highly prone to changes in temperature, humidity, wind speed, light intensity, and habitat structure, including disturbances [12–15]. In general, butterflies show a significant positive association with warm and dry summers [16], and their flights are affected by the cloud cover and wind velocity [17]. Different disturbance regimes, such as forest fires, grazing, and wood collection, also affect butterfly diversity, ultimately changing their habitat structure and community composition [18].

There are approximately 18,000 described butterfly species in the world [19], with about 90% found in the tropics [20], making the tropical region an important area for butterfly conservation [21]. Butterflies frequently need resources available in healthy ecosystems, such as flowers for nectar; specific host plants that caterpillars can use as a food source; and bare, moist soil patches for water and salts [22]. Butterflies help in pollination and support terrestrial ecosystem processes [23]. Butterflies are highly valuable indicator organisms. Butterfly population studies have greatly helped in assessing and restoring ecosystems' health [24]. Several studies have identified the significance of indicator species [25–27], where multispecies groups have proved to be good indicators of biotope quality. This study focused on butterflies of tropical dry forests and their correlations with microhabitat conditions to assess their ecosystem sustainability.

The Eastern Ghats, a region rich in biodiversity, provide immense opportunities for understanding butterfly populations in tropical dry forests. The discontinuous hill ranges of the Eastern Ghats support over 3200 angiosperms [28], 100 species of mammals, 425 species of birds, 199 species of herpetofauna, and 155 species of fish [29]. Over the last century, the Eastern Ghats have lost 15.83% of its forest cover and 34.45% of its core area [30]. Subsequently, suitable habitats for rare, endangered, and threatened species have been reduced by 0.08% [30]. The leading causes of this loss in forest cover included the expansion of agriculture and settlements, shifting cultivation, forest fires, indiscriminate logging, mining, construction of multipurpose dams, and infrastructure development [30,31]. In this study, we hypothesized that butterfly diversity is likely to be affected by the microclimate and anthropogenic disturbances, and we used human-impact signs, grazing signs, fire signs, and logging signs to evaluate the impact of disturbances on butterfly diversity. We considered the canopy cover, light intensity, relative humidity, and temperature to quantify the influence of the microclimate on butterfly richness. We also anticipated that variations in elevation and microhabitat, such as in open forests, riparian forests, and dense forests, would have an impact on butterfly diversity on a local scale, and that higher levels of disturbance together with higher light intensities and temperatures would negatively impact butterfly richness. Although butterflies may prefer riparian forests with a moderate daytime climate and availability of food and water, some butterflies showed a greater preference for open or dense forests. Rarefaction and extrapolation approaches, using Hill's numbers, were used to understand the species diversity and sample coverage in the different microhabitats, and structural equation modeling and canonical correspondence analysis were employed to quantify the causal effects of the candidate variables on butterfly diversity. Studying butterfly populations in the tropical dry forests of the Eastern Ghats is significant for the prudent management and conservation of butterflies.

## 2. Materials and Methods

### 2.1. Study Area and Sampling

We conducted this study in the northern parts of the Eastern Ghats hill ranges of the Koraput District, Odisha (17.4 to 20.7° N latitude and 81.24 to 84.2° E longitude), covering an area of 8807 km$^2$. The study area has distinct dry (October to May) and wet seasons (June to September) (Figure 1). Koraput has an elevation varying between 123 and 1655 m above mean sea level. The climate is sub-tropical with mean maximum and mini-

mum temperatures of 30.6 and 17.0 °C, respectively [32]. The mean annual precipitation is 1540 mm, and heavy downpours take place during the wet season [33]. The study area was divided into 2 × 2 km² grids [34,35], and 10% of these grids, i.e., 35 grids, were randomly selected for sampling [35] (Figure 2). We sampled 8 grids with dense forests (DF; canopy cover > 35%), 7 grids with riparian forests (RF; canopy cover between 18 and 52%), and 20 grids with open forests (OF; canopy cover < 35%), based on their dominance and availability (Table S1). We followed Pollard and Yates's standard butterfly counting protocol [17]. Butterflies were counted from 0900 to 1300 h via a transect walk method, with a constant space of 60 min for each transect. A transect of 500 m was surveyed in each sampling site (2 × 2 km² grids), with an average intervening distance of 12 km between grids, so that each transect represented an independent sample [36]. We counted butterflies for a six-month period during the dry season from December 15 to June 5 in the year 2019, as counting could not be undertaken during the wet season due to the cloudy skies and heavy downpours. The dry period supported high species richness, along with the possibility of observing anthropogenic disturbances. We followed taxonomy keys by Kehimkar [37] and Wynter-Blyth [38] for the identification of butterflies, as well as nomenclature by Varshney and Smetacek [39].

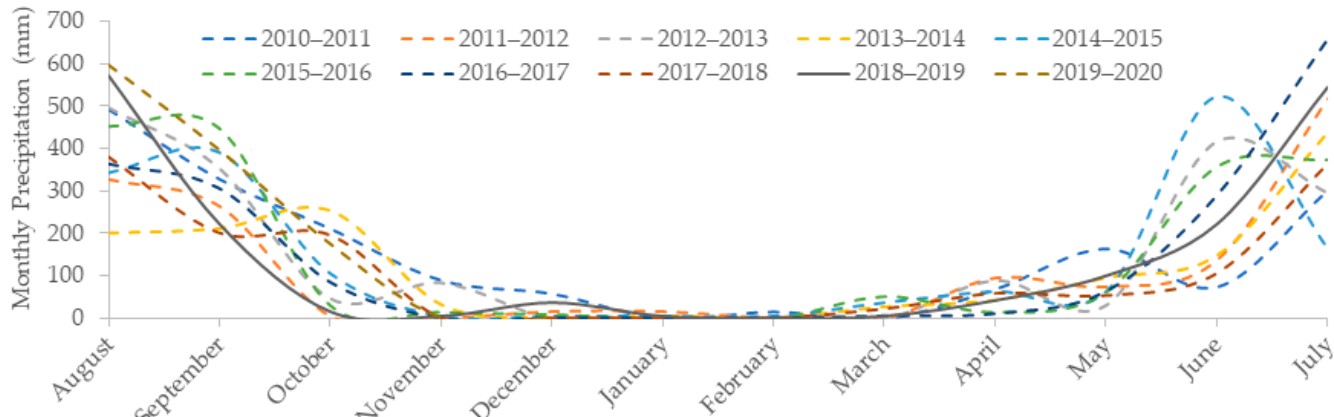

**Figure 1.** Monthly precipitation from August 2010 to July 2020 at Koraput, Odisha, India [40]. Fieldwork was conducted between 15 January and 15 June 2019. Precipitation from August 2018 to July 2019 is shown in a solid line.

*2.2. Vegetation Sampling*

Tree density (TD) was recorded in each of the sampling grids within a 5 m radius at 100 m intervals for each of the 500 m transects, within which a 2.5 m radius circular plot was used to quantify the shrub density (SD). Additionally, 2 circles of 1 m radius were used to sample the herb density (HD) 1 m away from the center of each segment [41]. The study considered trees to be above 10 cm diameter at breast height (1.37 m) for the calculation of TD. We considered plants with soft stems and height < 1 m as herbs and plants < 3 m in height and woody stems as shrubs [42]. Canopy cover (CC) was recorded in each 100 m section using GLAMA (Gap Light Analysis Mobile App. Version 3.0, Masaryk University, Brno, Czech Republic), an application in a smartphone with a 16 megapixel inbuilt camera held at breast height [43–45].

*2.3. Environmental Variable Measurements*

The study recorded elevation (ELV), temperature (TMP), relative humidity (RH), wind speed (WS), and relative light intensity (LI) in each 100 m section of each transect during the butterfly sampling period (from 0900 to 1300 h) and calculated the average values. Elevations were recorded using a GPS (GPSMAP 64s, Garmin, Olathe, KS, USA). Temperature, relative humidity, and wind speed were measured using a digital anemometer (AVM-06, HTC, Mumbai, India), and relative light intensity was measured using a digital

light meter (LX-103, Lutron, Gurgaon, India). We recorded logging signs (LS; number of logged trees), fire signs (FS; number of signs of past fires), livestock grazing signs (GS; number of livestock observed), and human signs (HS; number of humans observed using the transect as regular paths) along the 500 × 5 m transects by counting the total number of signs for each category separately (Table S1).

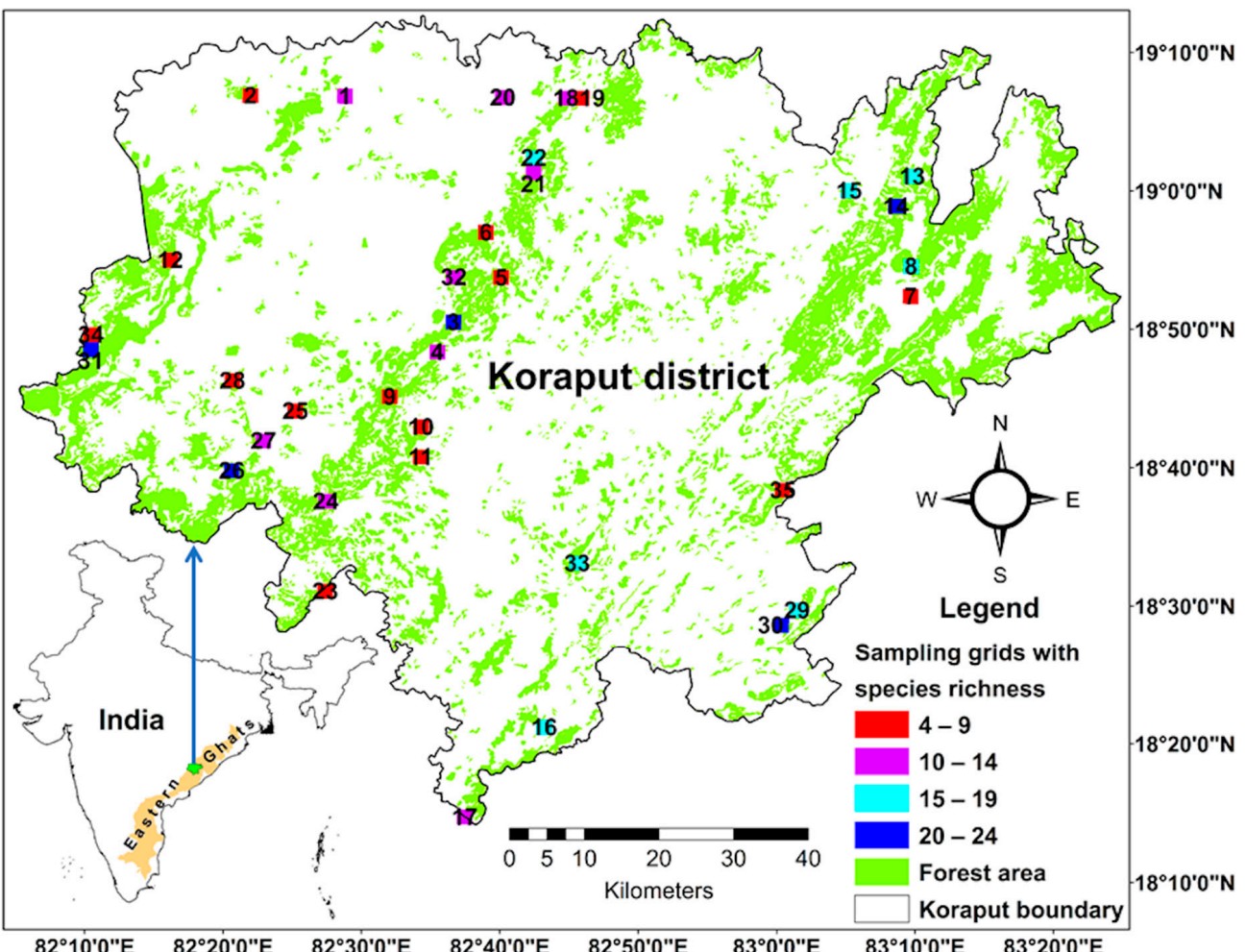

**Figure 2.** Study area map showing the sampling grids with grid numbers and species richness given in respective color codes.

### 2.4. Statistical Analysis

We quantified relative abundance as a percentage, representing the abundance of a species over the total number of individuals of all species. Habitat specialist species are distinguished by their association with particular habitats. The habitat specificity of a butterfly species was calculated by using the habitat specificity index (Sm), which represents the proportion of individuals in a preferred habitat to the total number of individuals [46]. Species with Sm values ≥ 0.9 were designated as habitat specialist species, Sm values < 0.5 were designated as generalist species, and 0.5 ≤ Sm < 0.9 values were designated as species with habitat preference (SHP) [46,47]. To examine the significance of habitat specialization for each species in the three forest types, a chi-squared ($\chi^2$) test with 1000 Monte Carlo simulations was performed [48]:

$$\chi^2 = \sum_{i=1}^{n} \frac{(O_i - E_i)^2}{E_i}$$

where $E_i$ and $O_i$ are the estimated and observed numbers of species in the *i*-th forest type, and *n* is the total number of forest type categories. The value of $E_i$ was obtained by multiplying the total number of butterflies of a focus species by the proportion of grids of the *i*-th forest type in the study area. The observed value was considered statistically significant ($p < 0.05$) if it fell within the top 5% of $\chi^2$ values produced by Monte Carlo simulations. In this work, a species with a significant habitat association is referred to as a "specialist species", whereas a species with a nonsignificant habitat association is referred to as either a "generalist species" or "species with habitat preferences" [48,49].

We identified indicator species based on the relationship of a species with one or more sites (hereafter called "site groups"), based on their environmental similarities, i.e., habitat types [50]. "Indicator value" indices were estimated as the probability of occurrence of species in these site groups. Our study used specific niche preferences of indicator species to detect the direction of changes in natural habitats; a moderate indicator value provides information about the direction of ecological changes [51,52]. We used the "indicspecies" package in R to identify the association of each species, or groups of species, to a specific microhabitat [52].

We used the rarefaction and extrapolation approach to assess the butterfly diversity in different forest types, based on the reference sample using Hill's numbers: species richness (q = 0), Shannon index (q = 1), and inverse Simpson index (q = 2) [53–56]. We performed extrapolation using the package "iNEXT" in R [56] for a number of survey sites (sample size *n*) that were double the size of the reference sample. Further, we obtained rarefaction/extrapolation biodiversity curves and sample coverage (SC) using the package "iNEXT" [56]. The structural equation model was applied using the "lavaan" package [57], and path diagrams were generated using the "SemPlot" package [58]. We compared the results of the structural equation model by canonical correspondence analysis [59] using the "vegan" package [60] and the three-dimensional ordiplots using the "vegan3d" package in R [61,62].

## 3. Results

In this study, 1614 individual butterflies were recorded in the study site, which included 79 species under 57 genera, 18 subfamilies, and 6 families (Table 1). The family Nymphalidae had the highest species richness (31), followed by Lycaenidae (22), Papilionidae (11), Pieridae (8), Hesperiidae (6), and Riodinidae (1). Open forests represented the highest number of butterfly species (58), followed by 56 in riparian forests and 46 in dense forests. Open and dense forests followed a similar family composition in species richness. The butterfly species of the riparian forests showed a different pattern: Nymphalidae > Lycaenidae > Pieridae > Papilionidae > Hesperiidae > Riodinidae. We recorded twenty-nine species (36.7%) that were common to all three forest types, of which only seven species were generalists, based on the habitat specificity index (Figure 3). Fourteen species were recorded exclusively in riparian forests, ten in open forests, and three in dense forests (Figure 3).

All specialist species were exclusive to the respective forest types except *Arhopalini amantes* and *Ariadne merione* (Table 1). The chi-squared ($\chi^2$) test for habitat specificity showed a nonsignificant result for both "generalist species" and "species with habitat preferences". In this study, most of the habitat specialist species were restricted to a single forest type, except *Arhopalini amantes* and *Ariadne merione*, which showed significant associations with open forests ($p < 0.05$). However, *Catopsilia pyranthe* and *Papilio demoleus* showed significant associations with a particular forest type ($p < 0.01$).

The butterfly richness along the elevation gradient showed an increasing pattern, followed by an undulating pattern, and the maximum species richness was observed at 750 m. The elevation range between 750 and 800 m was species rich, contributing 60.75% of the total number of recorded species. The study recorded two schedule I species (*Castalius rosimon* and *Neptis jumbah*), four schedule II species (*Cepora nerissa*, *Charaxes bernardus*, *Cyrestis thyodamas*, and *Lampides boeticus*), and one schedule IV species (*Euploea core*), under

the Indian Wildlife Protection Act of 1972. Butterfly species *Euploea core*, *Eurema hecabe*, *Junonia almanac*, and *Vanessa cardui* were included under the Least Concern (LC) Categories in the IUCN Red List (Table 1).

**Table 1.** List of butterfly species recorded from the study site (Koraput, Odisha) showing their habitat specificity index (Sm), measured as a proportion of the number of individuals in the preferred habitat to the total number of individuals. Sm ≥ 0.9 indicates habitat specialists (RFS: riparian forest specialist, DFS: dense forest specialist, OFS: open forest specialist), 0.5 ≤ Sm < 0.9 indicates species with habitat preference (SHP), and Sm < 0.5 indicates generalists [46,47]. Significant associations of habitat with the respective forest types were tested via a chi-squared ($\chi^2$) test (* and ** represent significance levels at 5 and 1%, respectively).

| Scientific Name | Common Name | Habitat Specificity Index | Habitat Specificity | Relative Abundance | | |
|---|---|---|---|---|---|---|
| | | | | Open Forests | Riparian Forests | Dense Forests |
| Family: Hesperiidae | | | | | | |
| Subfamily: Coeliadinae | | | | | | |
| *Hasora chromus* Subfamily: Hesperiinae | Common Banded Awl | 0.806 | SHP | 0.24 | 0.99 | 6.46 |
| *Iambrix salsala* Subfamily: Pyrginae | Chestnut Bob | 0.80 | SHP | 0.12 | 0 | 1.03 |
| *Sarangesa purendra* | Spotted Small Flat | 1 | RFS | 0 | 0.50 | 0 |
| *Sarangesa dasahara* | Common Small Flat | 1 | RFS | 0 | 0.25 | 0 |
| *Tagiades japetus* | Suffused Snow Flat | 1 | RFS | 0 | 0.25 | 0 |
| *Tagiades litigiosa* Family: Lycaenidae | Water Snow Flat | 1 | DFS | 0 | 0 | 0.26 |
| Subfamily: Curetinae | | | | | | |
| *Curetis acuta* Subfamily: Polyommatinae | Angled Sunbeam | 1 | RFS | 0 | 0.25 | 0 |
| *Acytolepis puspa* | Common Hedge Blue | 1 | OFS | 0.12 | 0 | 0 |
| *Anthene emolus* | Ciliate Blue | 1 | RFS | 0 | 0.25 | 0 |
| *Caleta decidia* | Angled Pierrot | 0.417 | Generalist | 0.61 | 0.99 | 0.78 |
| *Castalius rosimon* | Common Pierrot | 0.8 | SHP | 0.12 | 0.99 | 0 |
| *Catochrysops strabo* | Forget-Me-Not | 1 | OFS | 0.85 | 0 | 0 |
| *Chilades pandava* | Plains Cupid | 0.444 | Generalist | 0.36 | 0.99 | 0.52 |
| *Freyeria trochylus* | Grass Jewel | 1 | OFS | 0.49 | 0 | 0 |
| *Jamides bochus* | Dark Cerulean | 0.768657 | SHP | 2.67 | 2.23 | 26.61 |
| *Jamides celeno* | Common Cerulean | 0.534884 | SHP | 5.58 | 5.71 | 4.39 |
| *Lampides boeticus* | Peablue | 1 | OFS | 0.12 | 0 | 0 |
| *Neopithecops zalmora* | Quaker | 1 | RFS | 0 | 0.25 | 0 |
| *Prosotas nora* | Common Lineblue | 1 | RFS | 0 | 0.25 | 0 |
| *Pseudozizeeria maha* | Pale Grass Blue | 0.606 | SHP | 2.43 | 0.99 | 2.33 |
| *Zizeeria karsandra* | Dark Grass Blue | 0.4 | Generalist | 0.24 | 0.25 | 0.52 |
| *Zizula hylax* Subfamily: Theclinae | Tiny Grass Blue | 0.517 | SHP | 1.70 | 3.72 | 0 |
| *Amblypodia anita* | Purple Leaf Blue | 0.857 | SHP | 0.73 | 0.25 | 0 |
| *Arhopalini amantes* | Large Oakblue | 0.939 | OFS * | 5.58 | 0.74 | 0 |
| *Loxura atymnus* | Yam Fly | 0.5 | SHP | 0.24 | 0.74 | 1.29 |
| *Spindasis syama* | Club Silverline | 0.667 | SHP | 0.49 | 0 | 0.52 |
| *Spindasis vulcanus* | Common Silverline | 0.75 | SHP | 0.36 | 0 | 0.26 |
| *Zeltus amasa* Family: Nymphalidae | Fluffy Tit | 1 | RFS | 0 | 0.50 | 0 |
| Subfamily: Biblidinae | | | | | | |

**Table 1.** *Cont.*

| Scientific Name | Common Name | Habitat Specificity Index | Habitat Specificity | Relative Abundance | | |
|---|---|---|---|---|---|---|
| | | | | Open Forests | Riparian Forests | Dense Forests |
| *Ariadne ariadne* | Angled Castor | 0.667 | SHP | 0.49 | 0.50 | 0 |
| *Ariadne merione* | Common Castor | 0.9 | OFS * | 1.09 | 0.25 | 0 |
| Subfamily: Charaxinae | | | | | | |
| *Charaxes bernardus* | Tawny Rajah | 1 | RFS | 0 | 0.25 | 0 |
| *Polyura athamas* | Common Nawab | 1 | OFS | 0.24 | 0 | 0 |
| Subfamily: Cyrestinae | | | | | | |
| *Cyrestis thyodamas* | Common Map | 0.667 | SHP | 0.24 | 0 | 0.26 |
| Subfamily: Danainae | | | | | | |
| *Danaus chrysippus* | Plain Tiger | 0.694 | SHP | 3.03 | 2.23 | 0.52 |
| *Danaus genutia* | Common/Striped Tiger | 0.458 | Generalist | 1.33 | 1.74 | 1.55 |
| *Euploea core* | Common Crow | 0.398 | Generalist | 9.22 | 15.14 | 13.95 |
| *Parantica aglea* | Glassy Tiger | 0.538 | SHP | 0.36 | 1.74 | 0.76 |
| *Tirumala limniace* | Blue Tiger | 1 | RFS | 0 | 0.50 | 0 |
| Subfamily: Heliconiinae | | | | | | |
| *Acraea violae* | Tawny Coster | 0.60 | SHP | 0.24 | 0 | 0.78 |
| *Phalanta phalantha* | Common Leopard | 0.69 | SHP | 1.33 | 0.99 | 0.26 |
| *Vagrans egista* | Vagrant | 0.50 | SHP | 0.12 | 0.25 | 0 |
| Subfamily: Limenitidinae | | | | | | |
| *Athyma perius* | Common Sergeant | 1 | RFS | 0 | 0.25 | 0 |
| *Euthalia aconthea* | Common Baron | 1 | OFS | 0.24 | 0 | 0 |
| *Euthalia nais* | Baronet | 0.46 | Generalist | 1.33 | 2.48 | 0.78 |
| *Neptis hylas* | Common Sailer | 0.55 | SHP | 1.33 | 1.49 | 0.78 |
| *Neptis jumbah* | Chestnut-Streake Sailer | 0.75 | SHP | 0.12 | 0 | 0.78 |
| *Pantoporia hordonia* | Common Lascar | 0.50 | SHP | 0.12 | 0 | 0.26 |
| *Tanaecia lepidea* | Grey Count | 0.50 | SHP | 0 | 0.25 | 0.26 |
| Subfamily: Nymphalinae | | | | | | |
| *Hypolimnus bolina* | Great Eggfly | 0.67 | SHP | 0 | 0.50 | 0.26 |
| *Junonia almana* | Peacock Pansy | 0.50 | SHP | 0.36 | 0.25 | 0.52 |
| *Junonia atlites* | Grey Pansy | 1 | RFS | 0 | 1.24 | 0 |
| *Junonia iphita* | Chocolate Pansy | 0.43 | Generalist | 3.28 | 3.47 | 5.68 |
| *Junonia lemonias* | Lemon Pansy | 0.74 | SHP | 4.49 | 2.48 | 0.78 |
| *Junonia orithiya* | Blue Pansy | 1 | OFS | 0.12 | 0 | 0 |
| *Kallima inachus* | Orange Oakleaf | 0.80 | SHP | 0 | 0.99 | 0.26 |
| *Vanessa cardui* | Painted lady | 1 | DFS | 0 | 0 | 0.26 |
| Subfamily: Satyrinae | | | | | | |
| *Melanitis leda* | Common Evening Brown | 0.50 | SHP | 0.49 | 0.50 | 1.55 |
| *Mycalesis perseus* | Common Bushbrown | 0.69 | SHP | 0.24 | 2.23 | 0.52 |
| *Ypthima huebneri* | Common Fourring | 0.52 | SHP | 1.82 | 3.47 | 0 |
| Family: Papilionidae | | | | | | |
| Subfamily: Papilioninae | | | | | | |
| *Atrophaneura aristolochiae* | Common Rose | 0.50 | SHP | 0.12 | 0.25 | 0.52 |
| *Graphium agamemnon* | Tailed Jay | 0.67 | SHP | 0.24 | 0 | 0.26 |
| *Graphium doson* | Common Jay | 1 | OFS | 0.49 | 0 | 0 |
| *Graphium nomius* | Spot Swordtail | 0.78 | SHP | 2.55 | 0.99 | 0.52 |
| *Papilio clytia* | Common Mime | 0.5 | SHP | 0.36 | 0 | 0.78 |
| *Papilio crino* | Common Banded Peacock | 1 | OFS | 0.24 | 0 | 0 |
| *Papilio demoleus* | Lime Butterfly | 0.66 | SHP ** | 14.56 | 12.6 | 2.58 |
| *Papilio hector* | Crimson Rose | 1 | DFS | 0 | 0 | 0.26 |
| *Papilio paris* | Paris Peacock | 1 | OFS | 0.24 | 0 | 0 |
| *Papilio polymnestor* | Blue Mormon | 0.89 | SHP | 0.12 | 1.99 | 0 |
| *Papilio polytes* | Common Mormon | 0.52 | SHP | 3.40 | 2.48 | 4.13 |
| Family: Pieridae | | | | | | |

Table 1. *Cont.*

| Scientific Name | Common Name | Habitat Specificity Index | Habitat Specificity | Relative Abundance | | |
|---|---|---|---|---|---|---|
| | | | | Open Forests | Riparian Forests | Dense Forests |
| Subfamily: Coliadinae | | | | | | |
| *Catopsilia pomona* | Common Emigrant | 0.51 | SHP | 3.28 | 8.44 | 1.55 |
| *Catopsilia pyranthe* | Mottled Emigrant | 0.60 | SHP ** | 10.19 | 3.23 | 11.11 |
| *Eurema hecabe* | Common Grass Yellow | 0.69 | SHP | 4.37 | 3.72 | 0.26 |
| *Gandaca harina* | Tree Yellow | 0.82 | SHP | 2.18 | 0.74 | 0.26 |
| Subfamily: Pierinae | | | | | | |
| *Cepora nerissa* | Common Gull | 0.50 | SHP | 0.24 | 0.25 | 0.26 |
| *Delias hyparete* | Painted Jezabel | 0.88 | SHP | 2.67 | 0 | 0.78 |
| *Ixias pyrene* | Yellow Orange Tip | 1 | RFS | 0 | 0.25 | 0 |
| *Leptosia nina* | Psyche | 1 | RFS | 0 | 0.50 | 0 |
| Family: Riodinidae | | | | | | |
| Subfamily: Riodininae | | | | | | |
| *Abisara echerius* | Plum Judy | 0.8 | SHP | 0 | 0.25 | 1.03 |

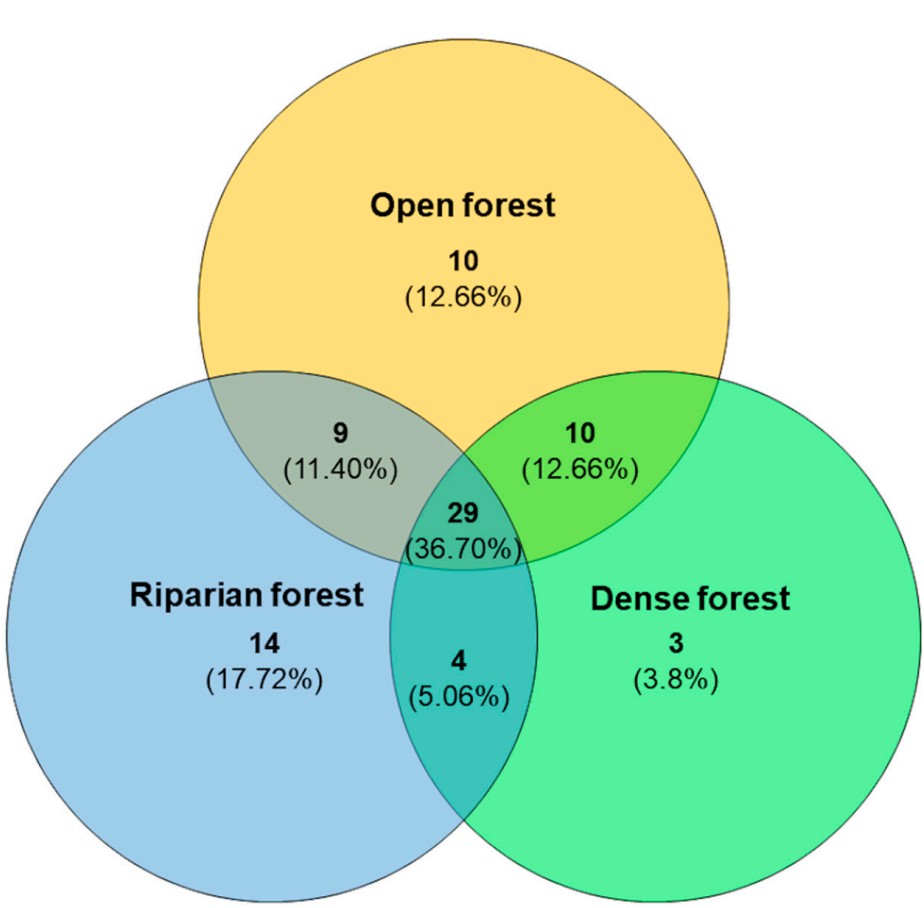

**Figure 3.** Venn diagram showing species richness (%) based on forest type.

The relative abundance of species was highest for the Nymphalidae family (35.39%), followed by Lycaenidae (25.28%), Pieridae (19.33%), Papilionidae (18.15%), Hesperiidae (2.54%), and least for the Riodinidae family (0.31%) (Table 1). The highest abundance per transect was recorded in riparian forests (57.57), followed by dense forests (48.38), and lowest in open forests (41.20) (Table S1).

The highest values of the Shannon index ($^1d$ = 45.47) and inverse Simpson index ($^2d$ = 36.39) in the reference sample were found for RF, with 56 taxa, followed by DF ($^0d$ = 46, $^1d$ = 38.23, $^2d$ = 32.38), then OF ($^0d$ = 58, $^1d$ = 37.78, $^2d$ = 27.82). When comparing the different forest types, OF showed the best sample completeness because, after an early rising period, the $^0d$ curve approached the horizontal. This indicates that the number of observed species did not rise after the initial period. The RF curve, on the other hand, showed a steady growth beyond the early period. This indicates that with more survey locations, new species are likely to be found (Figure 4). The sample coverage also supported this. OF showed the highest sample completeness, with SC = 92.6%, followed by DF (SC = 84.5%), then RF (SC = 72.9%) at q = 0.

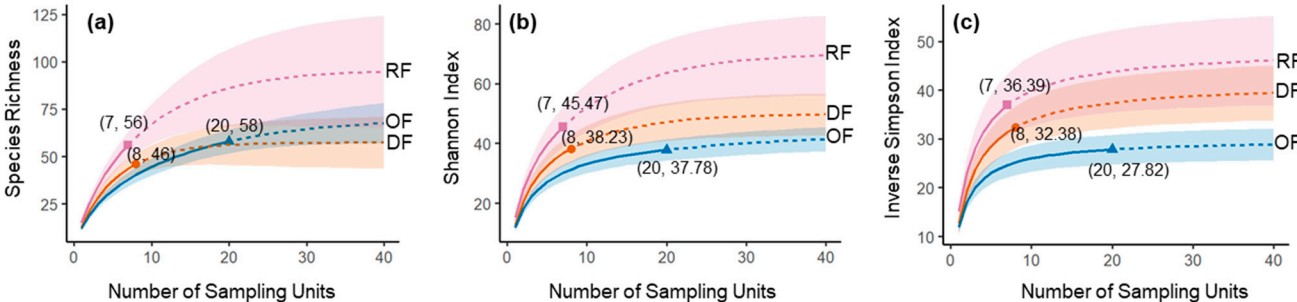

**Figure 4.** Rarefaction/extrapolation curves showing (**a**) species richness (q = 0), (**b**) Shannon index (q = 1), and (**c**) inverse Simpson index (q = 2). The solid line represents the rarefaction curve, and the extrapolation curve, which extends up to twice the size of the reference sample, is represented by the dotted line. The sample size and observed Hill's number in the reference sample are in parentheses, whereas points denoting biodiversity coordinates for the reference data are shown as dashed lines. The shaded areas represent the 95% confidence intervals derived using a bootstrap method for 50 replications. Habitat types include dense forest (DF), open forest (OF), and riparian forest (RF).

The indicator value analysis showed that none of the species solely acted as an indicator species for a specific forest type. The open and riparian forests showed combinations of species that served as indicators for their forest types. Although the combination of *Junonia iphita–Eurema hecabe* had a moderate "B" value of 0.57 for riparian forests, an indicator value of "A" equal to "1" was obtained for the following eight groups: *Caleta decidia–Eurema hecabe*, *Castalius rosimon–Catopsilia pomona*, *Arhopalini amantes–Mycalesis perseus*, *Arhopalini amantes–Parantica aglea*, *Zizula hylax–Parantica aglea*, *Mycalesis perseus–Ypthima huebneri*, *Phalanta phalantha–Neptis hyla*, and *Phalanta phalantha–Graphium nomius*. Alternately, two groups of species combinations served as indicators in dense forests: *Hasora chromus–Pseudozizeeria maha* and *Pseudozizeeria maha–Melanitis leda*, with "A" values of "1". *Hasora chromus* was shown to have a moderate "B" value of 0.5, indicating the ecology of dense forests. None of the species groups in the open forests satisfied the criterion for being an indicator species at *p*-values of 0.05 or 0.01 (Table 2).

The structural equation model showed that relative humidity had a negative correlation with elevation (37%), temperature (52%), and light intensity (54%). In contrast, relative humidity showed positive relationships with canopy cover (12%). Canopy cover was found to have a negative correlation with temperature (16%) and light intensity (14%), but it exhibited a positive correlation with elevation (11%). Elevation exhibited a negative correlation with temperature (27%) and showed a weak correlation with light intensity. Light intensity and temperature were found to have a strong positive correlation (60%). The model predicted a negative impact of light intensity (14%) on butterfly richness, which, in contrast, explained the positive influences of relative humidity (22%), elevation (16%), and temperature (40%) on richness. Canopy cover was found to have no significant relation to butterfly richness (Figure 5).

**Table 2.** Indicator value analysis for species combinations from three different forest types in Koraput. A = P(S/G) is the probability that the surveyed site belongs to the target site group G, given the fact that species S has been found, and B = P(S/G) is the probability of finding species S in sites belonging to the site group G. Statistical significance was tested using a permutation test (Stat), which involved comparing an observed test statistic with a distribution of the same statistic obtained by randomly reordering the data [63]. * and ** indicate 5 and 1% levels of significance, respectively.

| Species Combinations | A | B | Stat | *p*-Value |
|---|---|---|---|---|
| **Riparian Forests (No. of Species = 18)** | | | | |
| *Jamides celeno + Eurema hecabe* | 0.713 | 0.8571 | 0.782 | 0.003 ** |
| *Eurema hecabe* | 0.5268 | 0.8571 | 0.672 | 0.045 * |
| *Euploea core + Eurema hecabe* | 0.6202 | 0.7143 | 0.666 | 0.023 * |
| *Junonia iphita + Eurema hecabe* | 0.7004 | 0.5714 | 0.633 | 0.030 * |
| *Phalanta phalantha + Catopsilia pomona* | 0.8108 | 0.4286 | 0.589 | 0.038 * |
| *Junonia iphita + Mycalesis perseus* | 0.7921 | 0.4286 | 0.583 | 0.021 * |
| *Mycalesis perseus* | 0.786 | 0.4286 | 0.58 | 0.049 * |
| *Mycalesis perseus + Euploea core* | 0.7605 | 0.4286 | 0.571 | 0.029 * |
| *Caleta decidia + Eurema hecabe* | 1 | 0.2857 | 0.535 | 0.044 * |
| *Castalius rosimon + Catopsilia pomona* | 1 | 0.2857 | 0.535 | 0.038 * |
| *Arhopalini amantes + Mycalesis perseus* | 1 | 0.2857 | 0.535 | 0.030 * |
| *Arhopalini amantes + Parantica aglea* | 1 | 0.2857 | 0.535 | 0.030 * |
| *Zizula hylax + Parantica aglea* | 1 | 0.2857 | 0.535 | 0.033 * |
| *Mycalesis perseus + Ypthima huebneri* | 1 | 0.2857 | 0.535 | 0.033 * |
| *Phalanta phalantha + Neptis hylas* | 1 | 0.2857 | 0.535 | 0.032 * |
| *Phalanta phalantha + Graphium nomius* | 1 | 0.2857 | 0.535 | 0.033 * |
| *Ypthima huebneri + Parantica aglea* | 0.9449 | 0.2857 | 0.535 | 0.050 * |
| **Dense Forests (No. of Species = 8)** | | | | |
| *Hasora chromus + Catopsilia pyranthe* | 0.9756 | 0.5 | 0.698 | 0.006 ** |
| *Hasora chromus* | 0.8231 | 0.5 | 0.642 | 0.018 * |
| *Hasora chromus + Pseudozizeeria maha* | 1 | 0.375 | 0.612 | 0.009 ** |
| *Pseudozizeeria maha + Melanitis leda* | 1 | 0.375 | 0.612 | 0.016 * |
| *Melanitis leda + Papilio polytes* | 0.9259 | 0.375 | 0.589 | 0.04 |
| *Pseudozizeeria maha + Papilio polytes* | 0.6863 | 0.5 | 0.586 | 0.043 * |
| *Parantica aglea + Papilio polytes* | 0.8824 | 0.375 | 0.575 | 0.041 * |
| *Pseudozizeeria maha + Danaus genutia* | 0.7143 | 0.375 | 0.518 | 0.035 * |

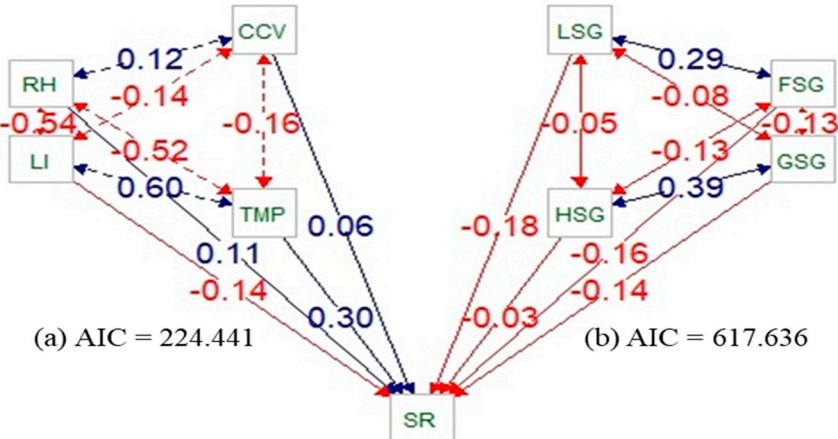

**Figure 5.** Path diagrams from the structural equation model representing correlations (dashed lines with arrows at both ends) and causative effects (single-headed arrows) of (**a**) the microclimate and (**b**) disturbances on butterfly richness (SR), where blue and red lines indicate positive and negative relationships between variables and butterfly richness, respectively. Abbreviations: light intensity (LI), relative humidity (RH), canopy cover (CCV), temperature (TMP), fire signs (FSG), logging signs (LSG), grazing signs (GSG), and human signs (HSG).

Fire signs and grazing signs were negatively correlated (13%). The correlations between logging signs and grazing signs, and logging signs and human signs, were nonsignificant. Disturbance parameters, primarily controlled by fire, grazing, and logging signs, showed negative impacts on butterfly richness (14 ± 4%) (Figure 5). Human signs exhibited nonsignificant effects on butterfly richness. The microclimate model was more precise (AIC = 224.441) than the disturbance model (AIC = 617.636).

The influences of the microclimate and disturbances on butterfly richness described by the structural equation model showed an overall agreement with the results of the canonical correspondence analysis (CCA). The first CCA axis, which explained 34% variance, was greatly associated with temperature, and the second axis was associated with relative humidity, with an explained variance of 29%. The light intensity was predicted to have greater significance in the third axis, with 20% explained variance. The disturbance parameters showed a similar pattern, with fire, grazing, and logging signs making dominant contributions to butterfly richness and being associated with the first and second axes, contributing 29 and 18% variance, respectively. Human signs were associated with the third axis (13% explained variance) and were less significant (Figure S1).

## 4. Discussion

Out of the 79 butterfly species found in the three different forest types, *Jamides celeno*, *Euploea core*, *Papilio demoleus*, and *Catopsilia pyranthe* were most dominant and contributed to 21.19% of the total records. The chi-squared ($\chi^2$) test for habitat specificity showed a significant association of *Papilio demoleus* and *Catopsilia pyranthe* species with particular forest types, showing habitat preferences. Their association with particular forest types may be due to the mud puddling behavior of *Papilio demoleus* and *Catopsilia pyranthe*, particularly in the riparian forest [64,65].

The relative humidity, light intensity, and temperature were key parameters influencing butterfly richness. This indicates that the microclimate has significant control over butterfly richness. The temperature was the most crucial factor, as it affected the activity, distribution, growth, oviposition, mating behavior, and larval development of butterflies [21]. The positive effects of relative humidity on butterfly richness explained that butterflies prefer warm and humid conditions, and an optimal humidity of 84–92% facilitated butterfly breeding [66]. Even though moderate temperature and humidity positively impacted butterfly richness, light intensity had a deleterious effect on their richness, suggesting that high radiation has a deleterious effect on the survival of butterflies [67,68]. The microclimate is significant for butterfly existence, and its association with vegetation structure makes it even more significant. The microclimate and vegetation structure greatly influence butterfly distribution in microhabitats [2]. The presence of vegetation significantly impacts butterfly distribution, due to being a source of food and shelter. Our results support the earlier findings of Ghazoul [69], who observed the dominance of butterfly species in a tropical dry forest in Thailand.

In addition to microclimate, disturbances have a key significance on butterfly distribution. Anthropogenic disturbance parameters, such as logging, fire, and grazing, negatively impacted butterfly richness. This indicates that anthropogenic disturbances are likely to affect the butterfly distribution. A suitable microclimate influences microhabitat, diapause, or larval growth, and food availability is significant for the survival of butterflies [5,70]. Vegetation structure shapes the microhabitat condition, which ultimately provides food and shelter to butterflies. Anthropogenic disturbances disrupt the natural ecosystem, and the availability of host plants and nutrients for butterflies becomes scarce. Our findings are consistent with earlier reports that showed climate or anthropogenic effects on canopy cover and vegetation density in tropical dry forests influenced the microclimate [6,7,71], which ultimately influenced the butterfly richness of a microhabitat [72]. Devries and Walla [73] also claimed that microclimatic factors and vegetation heterogeneity promote a more diverse, but patchy distribution of butterflies. The nonsignificant impact of wind speed on butterfly richness explains that the wind intensity of the study area was suitable

for their distribution. On the other hand, the availability of butterflies was greatest at elevations between 750 and 800 m, suggesting a mid-domain effect on species richness along an elevational gradient [74,75]. Due to similar reasons, the species richness in the eastern ranges, such as the Laxmipur Pottangi and Deomali Hill ranges, are higher than that of the western low elevation zone of the Jeypore plateau. Gallou et al. [76] also observed that butterfly richness increased along an elevational gradient, reaching a maximum at 700 m and then sharply decreasing at 1900 m.

The riparian forests had great species diversity and species uniformity compared with the open and dense forests. The riparian forests also had the greatest number of groups with eight indicator species combinations, suggesting prioritization of its conservation. The greater the number of multispecies groups present, the higher the requirement for conservation [77]. Dense forests had two groups of indicator species and open forests had no indicator species groups. This suggests that riparian forests have greater butterfly diversity than open and dense forests. An and Choi [78] reported that a riparian forest is a preferred habitat for butterfly diversity. The dominance of butterflies in riparian forests also explains the significance of the ecotone on greater species diversity. The suitable microclimate and food and water availability of riparian forests make butterflies prefer these habitats. Previous studies also support our results, showing that riparian forests have high butterfly diversity [79]. The open forest is a more preferred habitat than that of the dense forest, indicating that butterflies prefer a low canopy. Earlier studies also reported that open forests attract more butterfly species than natural dense forests [80,81]. Although dense and open forests differ in canopy cover, canopy cover had no direct influence on butterfly richness. However, canopy cover had an indirect effect on temperature, which greatly affected butterfly distribution. A moderate microclimate and fewer disturbances in riparian forests make it a preferred habitat than open and dense forests for butterflies. The ecological complexities of the riparian forests, which act as an ecological conduit for the wildlife and ecotones of different habitat fragments [10], are more suitable for butterfly diversity. In contrast, open forests are prone to disturbances and are mainly associated with a dominant species, where the generalist species reduce the species diversity of these degraded ecosystems. Forest types and their vegetation structure influence the butterfly distribution because of their host-specificity, food availability, and shelter [8]. The microclimate, water availability, and nearby host plants facilitate butterfly distribution. In contrast, anthropogenic disturbances reduce butterfly diversity. Our study supports earlier findings that open forests with disturbed habitat conditions and low understory vegetation were less diverse than dense forests [11]. Though dense forests support fewer butterfly species, increases in disturbances are likely to further reduce butterfly endemicity [41,82,83].

## 5. Conservation Prioritization

The negative effect of light intensity on butterfly richness is likely to increase with deforestation. Land use and land cover change create tremendous pressure on butterfly microhabitats. The microclimate and disturbances regulate butterfly distributions, where vegetation structure and forest type play significant roles. Though large areas of intact forest would ensure the preservation of tropical biodiversity, in most cases, this is not a viable option. Therefore, maintaining intact forest areas in a matrix of sympathetically managed production forests would be a more appropriate management strategy [84]. In the last five decades, the Eastern Ghats landscape and forests around Koraput have been anthropogenically altered on a large scale [85], indicating that the existing conservation strategy requires greater ecosystem monitoring and sustainable management. Sustainably managed conservation is significant for maintaining species diversity for the region, focusing exclusively on butterfly species for a specific habitat. Although riparian forests need greater attention for butterfly conservation, maintaining open and dense forests is also crucial in preserving each microhabitat's species diversity and endemicity. For a comprehensive understanding of the effects of the microclimate and disturbances on butterfly

distribution and their assemblages, we highly recommend long-term monitoring of the different forest types in the Eastern Ghats.

**Supplementary Materials:** The following supporting information can be downloaded at https://www.mdpi.com/article/10.3390/cli11110220/s1, Figure S1: Three-dimensional ordination diagram between butterfly richness and (a) microclimatic variables; (b) disturbance parameters, derived using canonical correspondence analysis. Abbreviations: light intensity (LI), relative humidity (RH), canopy cover (CCV), temperature (TMP), fire signs (FSG), logging signs (LSG), and grazing signs (GSG). Table S1: Grid locations, habitat types, and description statistics of the following parameters: elevation (ELV), canopy cover (CC), temperature (TMP), relative humidity (RH), light intensity (LI), wind speed (WS), human signs (HS), grazing signs (GS), logging signs (LS), and fire signs (FS).

**Author Contributions:** Conceptualization, S.K.P. and R.M.P.; methodology, S.K.P., R.M.P. and A.M.; software, R.M.P. and A.M.; validation, A.M., R.M.P., S.K.P. and P.D.; formal analysis, A.M. and R.M.P.; investigation, A.M., A.N., A.K.N. and S.K.P.; resources, S.K.P.; data curation, A.M., A.N., A.K.N. and R.M.P.; writing—original draft preparation, A.M. and R.M.P.; writing—review and editing, S.K.P., R.M.P. and P.D.; visualization, R.M.P.; supervision, S.K.P.; project administration, S.K.P.; funding acquisition, S.K.P. and P.D. All authors have read and agreed to the published version of the manuscript.

**Funding:** This work was supported by the University Grant Commission, New Delhi, India, with award number CUO/ACA/NNFPHD/135.

**Data Availability Statement:** All data generated or analyzed during this study are included in this published article in the form of figures and tables. Additional information about the dataset or accessing the dataset in a different format than what is presented in this article can be obtained from the corresponding authors upon request.

**Acknowledgments:** The authors are grateful to the University Grant Commission, New Delhi, India, for providing financial support for fieldwork. We are also thankful to the Koraput Forest Division, Koraput, Odisha, India, for the necessary support to carry out this study.

**Conflicts of Interest:** The authors declare no conflict of interest. The funders had no role in the design of the study; in the collection, analyses, or interpretation of data; in the writing of the manuscript; or in the decision to publish the results.

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
