# Peer review of "Microclimate and Vegetation Structure Significantly Affect Butterfly Assemblages in a Tropical Dry Forest"

_climate, doi:10.3390/cli11110220_

Round 1
Reviewer 1 Report
Comments and Suggestions for Authors
Microclimate and Vegetation Structure Significantly Affect the Butterfly Assemblages in the Tropical Dry Forest of Eastern Ghats of India.
The authors have read a large amount of literature and performed a field study at a large spatial scale, making observations on butterflies as well as a large number of (and types of) environmental factors. They used a large number of sophisticated statistical techniques. The main questions were about, first habitat specialization of species and, second on species richness patterns across habitats and how that relates to environmental factors (second question is in the title). The study site is in the Eastern Ghats of India which is a dry forest region. Both the region and dry tropical forests deserve more study. there are 3 habitats included in the study (contrary to the title). So this is a welcome contribution and I congratulate the authors with their achievement.
While the spatial setup is good, the temporal aspects (not well described) is not so good so that the final sample size is small (1614 individuals). All sites were only sampled during 1 dry season (probably once). In tropical systems, such a sample size and a single point in time of sampling is generally not enough to get a fair picture of the species diversity of a site. Populations fluctuate, some species may have a diapause phase or migrate away during the dry season. Nevertheless, it is possible to study the habitat specialization of those species with sufficient sample size and perhaps say something about the differences between the 3 habitats in butterfly community, at least for the dry season. We may predict that during the wet season, some of the species that stay in riverine gallery forest during the dry season may move out to the dryer habitats. So it should be made clear that the results are limited to the dry season.
The statistical approach has a similar flaw; just trying everything out. I generally have a problem with methods in which what other species do affect the ‘habitat specialization’ of a given species as detailed below (something I encounter often). To my mind, simply calculating proportions based on the number of individuals in each habitat is just fine. You can do Chi-square tests to see if the specialization is significant (use correction for multiple testing if that is useful). What is wrong with that? Why these other methods?
Can you look at how the environmental factors are related to the most common species?
Can you compare the proportion of specialists in each habitat?
To compare species richness of sites, I would go for rarefaction analysis rather than ‘estimate S approaches’. However, given the limited sampling, I am not convinced that pursuing this is useful.
Some specific comments:
Methods.
L111. Site (delete last s) . How ‘2km blocks but 12.5 km between transects’?
L113 1 season (delete s). Meaning of “and the possibility of anthropogenic disturbance during this period.”?
L114 -> identification keys. ‘scientific naming’ is part of the taxonomy (often called ‘nomenclature’).
L121 What is the meaning of the second part of “We disregarded trees below 10 cm diameter at breast height (1.37 m) over barks and dead trees having utility less than 70%.”
L129 over what period were these parameters recorded? Measuring e.g. wind speed on 1 day doesn’t make much sense.
L142 Why was the number of individuals of other species made to affect how a given species was (statistically) responding to micro-climate ‘Relative abundance was quantified as the percentage of each species to the total number of individuals of all species’?
I like the clear definition of habitat specialization and the classification. This seems to assume that there are always 2 habitats compared, but there are 3 in e.g. figure 2. Please explain more. Next, I am then confused by ‘Indicator species’: why is that not the same as ‘habitat specialist’? How exactly was indicator status calculated? Later you cite the R package, but what does that do? Generally, an ‘indicator species’ should indicate something that is otherwise more difficult to observe (e.g. pollution). Is it difficult to distinguish the 3 habitat types? I guess not. So I would focus on habitat specialization. This appears to be a common flaw but that is no reason to also do this.
Later in results table 1 I learned that ‘habitat specialist’ is based on the relative abundance rather than actual abundance. I don’t see why the habitat specialization of a species should be in any way influenced by what other species are doing. I see this a lot (such as in the polynomial analysis of the CLAM program) and have never got an explanation why this is useful. This does not work if total abundance differs between habitats, as is usually the case!
‘Habitat specialization’ and ‘Indicator species’ is basically the same thing so you should choose 1 way to calculate it. In results, there are now 2 tables (1 and 3) that are on this topic, separated by a section about the richness etc.
L150-153 are messy.
L157 This paragraph would be a first step in comparing species richness, diversity, or community-weighted means among places, but this is not the topic of the paper. What is the reason for these analyses?
To compare species richness it seems a better option would be to use rarefaction analysis (which includes error estimation), rather than to rely on extrapolation to guess the total number of species.
L160 to 162 doesn’t seem to make sense. The number of singletons is an observed number.
L167 What question does the structural equation model address? It seems a new topic for a new paragraph.
Results
L177 -> number of species
L178 what is ‘richness’? Species observed or estimated? Rarefaction analysis? Estimators were announced above but here ignored.
Table 2: the system of 2 / 5 etc is not nice. Just create separate columns for each parameter.
L231 The structural equation model. Did you have enough data to do this? Why are there 2 models? Is it not possible to exclude non-sensical relations such as human signs affecting wind speed? The effect of elevation should be via its effect on temperature (maybe windspeed) so putting them together in a model may not make sense. Some measures of model fit probably need to be provided. Which relationships are significant?
Can you look at how the environmental factors are related to the most common species?
Figure 4. 3D figures can only be useful when you can move them. Printing a 3D figure in 2D can only be used to deceive readers and should thus not be included.
Comments on the Quality of English Language
While the writing sounds good, it is poorly structured. Individual sentences are put together without much of a story (in the introduction and discussion). To improve this, authors are advised to first create topical sentences for each paragraph that together form the argument of the paper. Only when you have created such a clear outline in the internal discussion among authors, you can fill in the details. In the introduction, try to have paragraphs that have a conclusion that suggests gaps in our knowledge that your study will partly fill (last sentemnce of a paragraph). Once you have your own structure setup, you support it with citations. This prevents the various papers that you are reading from pushing you to write incoherent text based on bits of ‘true’ text. In this case, a fresh start with a blank slate may be most effective. Lots of good info is there, but it needs to be restructured.
Author Response
Please find our responses in the uploaded pdf.

Reviewer 2 Report
Comments and Suggestions for Authors
This study analyzed the factors that determine the distribution and diversity of butterflies in tropical dry forests in India, mainly in terms of climatic factors, vegetation structure, and human disturbance. Although the results obtained are not highly novel, as many previous studies have already clarified many of the issues, the study is worth publishing as a case study that provides more support for previous cases.
I believe that the overall tone of the study is solid and well described. However, the most important factors that define the distribution and community structure of butterflies are, of course, mainly the climatic and physical factors analyzed by the authors, but in parallel, the food resources of butterflies (host plants, nectar plants, etc.). Rather, the latter are the most important direct factors that define butterfly distribution and community structure. However, there is no mention of this in the present paper. Of course, it may not be possible to mention this point in the results, since these do not seem to have been obtained as data this time, but I think it is essential to mention it in the discussion.
In the discussion, it is mentioned that vegetation structure is an important factor affecting the distribution of butterflies, and I believe that vegetation structure, of course, greatly affects the distribution of butterflies' food resources, which directly defines the distribution and community structure of butterflies. I would like you to consider how the climatic and physical factors of the study site relate to vegetation and food resource distribution, and how they affect the distribution and community structure of butterflies in the study site. Of course, if you have data on food resources in the study area, I would appreciate your analysis and addition of such data, which would make the results obtained in this study more robust and would further enhance the publication value of this study.
Author Response

(The authors gave the same response as above.)

Reviewer 3 Report
Comments and Suggestions for Authors
An interesting article that deserves to be published. However, the text as a whole needs to be restructured and unnecessary repetition avoided to make it easier to read. I think the changes needed are purely to the form and not to the experimentation and treatments. My proposal is therefore "minor revisions".
Title: It may not be necessary to indicate the location. If your conclusions can be considered general for the dry tropical forest, the study site is accessory.
Abstract:
Line 14: in my mind remove “where the butterfly diversity studies are limited”. This information have to be provide in introduction eventually or even in Material and Methods.
The structure of the abstract needs to be modified, as some results come before the methods:
· Justification/context
· How was it done
· Results
· Main conclusions
Key words: here please place at least the country. Try as much as possible not to use words from the title (not always easy…)
Introduction: In general try to divide into paragraphs to make it easier to read.
Line 39: "larval growth" “diapause” The beginning of your introduction is general (that's OK) and all of a sudden you're talking about phytophagous larvae/diapause... so either you mention in this sentence that you're going to work on insects or you delete these words.
Line 42: You speak directly of butterflies, you’ve got to introduce “insects/bio-indicators” before.
Line 50: Your references are centered on north holarctic studies. There are a lot published by Brazilian authors on tropical dry forest (authors: Freitas, Freire, Agra) alternatively you may cite: Legal L, Valet M, Dorado O, Jesus-Almonte JM, Lopez K, Céréghino R (2020) Lepidoptera are relevant bioindicators of passive regeneration in tropical dry forests. Diversity-Basel. 12:231
Line 63: the reference mentioned just above can fit here.
Material and Methods:
Line 106: Really, chosen randomly? Most of points are situated on the road from Balimela to Boriguma.
Results: As with the introduction, please structure your text in paragraphs with subheadings to make it easier to read.
Line 176-177: I don't know India, but do these numbers (in proportion) make any sense given the known fauna? Very few species of Hesperiidae, is it normal?
Table 1: Please correct. Two times Charaxinae. The genus Ariadne belongs to Biblidinae.
There are many little wars concerning “Papilio” names. Better to put Chilasa clytia or to keep Papilio? (Please check the best, the two are correct for me).
Line 200: repetitive with lines 176-177.
Table 2: I had to search in Material and Methods to understand the difference between Singletons and Unique. For this last may be to change by Single/Several sites to be more explicit.
Figure 4 is not easy to read. A way to present more clearly these results?
Discussion: Once again text is relevant but poorly organized => difficult to read
In the "Conservation" section, it would be interesting to identify the species that can only be found in a specific type of forest and not in others, and to check whether these species are endemic to the region so that conservation efforts can focus on these priority species.
Comments on the Quality of English LanguageI'm not a native speaker, but I have the impression that some of the wording seems strange to me.
Author Response

(The authors gave the same response as above.)

Round 2
Reviewer 1 Report
Comments and Suggestions for Authors
The authors have attempted to address the reviewer's concerns to some extent. It is good that rarefaction analysis was included. However, the rarefaction should be used to compare habitats, so the configuration is confused.
I now understand that wind speed and temperature were measured during the surveys. Thus, they to a very limited extent characterize the habitats and rather the sampling conditions. Sampling was targeted to take place when the weather was good for butterflies: no wind and warm. Therefore, the weather (wind speed, temperature) should rather be a covariate and not a predictor in models. You could e.g. model the abundance of each species with respect to weather and then do your analyses of the residuals (removing wing speed). Often the weather at sampling is simply ignored in such analyses after taking care to avoid bad weather when sampling
Authors persist in allowing what other species do to affect estimates of habitat specialization. I recommend using raw values and do Chi2 tests. The tests were included in the table on the old analyses as *. Authors did not explain why what other species do should affect how we judge the habitat specialization of a given species.
Comments on the Quality of English LanguageThe text was revised superficially and still has poor structure (especially Introduction and Discussion) and remains full of sentences that have unclear meaning. Some polishing up using e.g. Grammarly might help a bit. A fresh rewrite was not done.
Author Response
The review report is attached.

Reviewer 2 Report
Comments and Suggestions for Authors
I understand that corrections to my comments were made in the discussion section, but unfortunately, reading the discussion section, it does not appear that much correction was made. However, I understand that a survey of butterfly food resources and their relationship to vegetation structure and microclimate was not conducted at this time, so I understand that significant discussion is not possible. I would like to agree with the publication of the important results of the relationship between physical distribution factors and butterfly community structure this time, in that I look forward to future surveys on these points.
Author Response
The review report is attached.
